# Procoagulant Activity of Umbilical Cord-Derived Mesenchymal Stromal Cells’ Extracellular Vesicles (MSC-EVs)

**DOI:** 10.3390/ijms24119216

**Published:** 2023-05-24

**Authors:** Adrienne Wright, Orman (Larry) Snyder, Hong He, Lane K. Christenson, Sherry Fleming, Mark L. Weiss

**Affiliations:** 1Department of Anatomy and Physiology, Kansas State University, Manhattan, KS 66506, USA; adrwright@kbibiopharma.com (A.W.); ljsnyder@ksu.edu (O.S.); hhe@vet.ksu.edu (H.H.); 2Midwest Institute of Comparative Stem Cell Biotechnology, Kansas State University, Manhattan, KS 66506, USA; 3Department of Cell Biology and Physiology, University of Kansas Medical Center, Kansas City, KS 66160, USA; lchristenson@kumc.edu; 4Division of Biology, Kansas State University, Manhattan, KS 66506, USA; sdflemin@ksu.edu

**Keywords:** exosome, hemocompatibility, mesenchymal stromal cells, clinical safety, tissue factor, canine, human

## Abstract

Many cell types, including cancer cells, release tissue factor (TF)-exposing extracellular vesicles (EVs). It is unknown whether MSC-EVs pose a thromboembolism risk due to TF expression. Knowing that MSCs express TF and are procoagulant, we hypothesize that MSC-EVs also might. Here, we examined the expression of TF and the procoagulant activity of MSC-EVs and the impact of EV isolation methods and cell culture expansion on EV yield, characterization, and potential risk using a design of experiments methodology. MSC-EVs were found to express TF and have procoagulant activity. Thus, when MSC-derived EVs are employed as a therapeutic agent, one might consider TF, procoagulant activity, and thromboembolism risk and take steps to prevent them.

## 1. Introduction

Mesenchymal stromal cells (MSCs) have been implicated as a therapeutic modality in immune-mediated diseases, such as Crohn’s and Graft vs. Host disease, because they stimulate tissue regeneration and have anti-inflammatory effects [1,2,3,4,5,6]. Although it was once believed that MSCs’ actions were mediated through engraftment and differentiation, the current thought is that their physiological actions are mediated by the secretion of bioactive soluble factors and the release of extracellular vesicles (EVs) [7,8,9,10]. While MSCs are not yet approved as cellular therapeutics in the USA, they are licensed in Japan, Canada, and Europe. MSCs are widely considered to be “safe”, but they may pose a thromboembolism risk due to their procoagulant activity or the expression of tissue factor (TF) on their surface [11,12,13,14,15,16].

Many cell types express tissue factor (TF), which is also known as CD142 or coagulation factor III. Tissue factor freely circulates in the blood in its soluble form [17,18,19,20], providing supplementary protection to organs susceptible to mechanical injury. Tissue factor activates the coagulation cascade’s extrinsic arm through its extracellular domain binding to factor VII and factor VIIa (FVIIa), creating a TF-FVIIa covalent complex [18,19]. Following TF-FVIIa complex formation, factors IX and X of the intrinsic and common pathways are activated, leading to clot generation via thrombin, platelet activation, and fibrin deposition [18,19,21].

MSCs may be functionally procoagulant when they are exposed to blood or plasma, and that activity is due to TF expression [11,12,16,22]. Oeller et al. demonstrated that MSCs derived from both the human umbilical cord (UC-MSCs) and adipose tissue (AD-MSCs) exhibit more TF expression compared to those derived from bone marrow (BM-MSCs) and that the level of TF expression increased due to cell culture medium supplementation [16]. Similarly, Christy et al. reported that AD-MSCs had higher levels of TF expression than BM-MSCs did, and the percentage of TF-positive AD-MSCs declined after 15–20 population doublings in cell culture. At the same time, the TF-positive BM-MSCs exhibited no cell culture expansion change (NB: MSCs will most likely not be clinically used beyond 15 population doublings). In contrast, the portion of TF-positive BM-MSCs has no expansion-related change [11]. Christy et al. demonstrated that the percentage of TF-positive MSCs roughly correlates with their functional procoagulant activity [11]. George et al. demonstrated that MSC procoagulant activity is at least partially dependent on TF expression since incubation with a TF-neutralizing antibody (clone TF8-5G9) caused the loss of MSC’s functional procoagulant activity [12]. Regarding the procoagulant risk, Tatsumi et al. showed that the intravenous administration of AD-MSCs results in ~85% mortality among mice within 24 h of transplantation due to the formation of a pulmonary embolism [22]. The procoagulant activity of mouse MSCs was inhibited using an anti-TF antibody or factor VII-deficient plasma [22]. In summary, TF expression and the procoagulant activity of MSCs may comprise a patient safety risk factor. The question addressed here is whether extracellular vesicles (EVs) isolated from an MSC-conditioned medium (CM) pose a risk or might be a safer alternative therapy to MSCs since some researchers consider EVs to be immunologically inert [23,24,25].

EVs are phospholipid bilayer membrane nanoparticles released by many cells, including MSCs [26,27,28,29,30]. These particles are grouped by size and the mechanism of cellular release into three subpopulations: apoptotic bodies (the largest ones), microvesicles (intermediate size), and exosomes (the smallest ones) [7,31]. Exosomes range in diameter from 30 to 150 nm and are released via the fusion of multivesicular bodies with the plasma membrane [27]. Microvesicles are, on average, larger than exosomes and are formed and released from cells using a mechanism that differs from that of exosomes [32]. EVs are involved in intercellular communication, signaling, antigen presentation, cell adhesion, inflammation, and tissue remodeling and may function as disease biomarkers [26,33,34,35,36,37,38]. EVs retain “artifacts” of the parental cell type, such as cargo and cell surface marker expressions [23,39]. The plasma membranes of exosomes and parental cells differ slightly since exosomes enrich endocytosed extracellular ligands, lipid raft proteins, and raft-associated lipids such as GM1 gangliosides [40,41,42,43]. Regarding the TF expression and procoagulant activity of EVs, TF-exposing EVs have been noted in the blood, urine, and saliva [44]. Several cell types release TF-exposing EVs, including activated monocytes, endothelial, and many cancer types [44,45,46]. Increased numbers of EVs, particularly TF-positive EVs, have been noted in patients with cancer, endotoxemia, and atrial fibrillation [47,48]. In addition, EVs derived from different tumor types have procoagulant activity [49,50,51,52,53]. However, to our knowledge, the TF expression or procoagulant activity of canine MSC-derived EVs is unknown. Since the dog is considered to be an excellent large animal model for human-inherited bleeding disorders and has contributed to understanding human tissue factor and Factor VIIa complex [54,55], we evaluated canine MSC-derived EVs for TF expression and procoagulant activity.

## 2. Results

### 2.1. EV Characterization

EVs were characterized per ISEV recommendations [56]: nanoparticle tracking analysis (NTA), transmission electron microscope (TEM), dynamic light scattering (DLS), protein concentration, and dot blotting (Appendix A). Additionally, EVs were characterized for TF expression and procoagulant activity (described below).

### 2.2. Nanoparticle Concentration via Nanoparticle Tracking Analysis (NTA)

The effect of cell passaging on nanoparticle concentration was observed in passages 2–12 (Appendix A), and no significant differences were observed via passaging. A trend of decreasing nanoparticle concentration with each passage was observed (Appendix A), and when “passages” were assigned to two epochs: early vs. late passages, as shown in Appendix A, the nanoparticle concentration between the “early passage” (P2–P5) and “late passage” (P9–P12) was significantly different. To examine whether rapid MSC expansion affected the nanoparticle concentration, we observed a weak association between the population doubling time and nanoparticle concentration (Appendix A). Notably, the isolation method strongly impacted the nanoparticle concentration, with UC producing an average (±SD) of 1.02 × 10^10^ ± 6.2 × 10^9^ nanoparticles/mL compared to 6.73 × 10^9^ ± 3.4 × 10^9^ for the SEC method, i.e., 33% more nanoparticles /mL following ultracentrifuge (UC) isolation (Appendix A). Appendix A indicates that UC isolation had more EV yield variability than SEC did. This difference was seen in the early and late passages (Appendix A). Similar trends were observed when they were plotted as nanoparticles released per MSC cell (Appendix A). The median of about 20,000 nanoparticles per canine MSC in early passages (Appendix A) was less than what we reported for human MSCs in the early passages following 24 h of medium conditioning [57].

### 2.3. Nanoparticle Size

EV size was evaluated via NTA, DLS, and TEM (Appendix A). All three methods placed EV size within the range of exosomes (e.g., 50–150 nm). The NTA modal size was compared between the early and late passages, and no differences were found (Appendix A). However, a trend of the size decreasing with the later passage was observed; EVs isolated from the early passage had a median size of 93.0 nm compared to 88.0 nm for late passage EVs. In contrast, the NTA-based sizes of EVs isolated via SEC vs. UC were significantly different (Appendix A), with EVs isolated via SEC having a larger size (95.3 nm compared to UC at 85.8 nm; ~11% larger). The hydrodynamic size of EVs estimated via DLS was compared between the early and late passages and the isolation method. As shown in Appendix A, the DLS-determined size was not different between EVs isolated from the early or late passage CM. Early passage EVs had a diameter of 155.1 nm, which was compared to 147.6 nm for late passage EVs. In contrast, as shown in Appendix A, EVs isolated via SEC were ~15% larger than those isolated using UC (151.2 nm vs. 128.4 nm, respectively. When the EV size was estimated using TEM (Appendix A), 31 to 40 EVs were measured per group. As depicted in Appendix A, EVs appeared to be roughly spherical and ranged in diameter from 38 nm to 218 nm. In TEM, a distinct black ring or doughnut structure was observed to indicate a bilayer structure. More small debris was observed in UC-isolated EVs than the number of those in SEC (Appendix A, top panel vs. bottom panel). As shown in Appendix A, in the top panel, size differences were observed in TEM when early passage (P2–5) and late passage (P9–12) EVs were compared. EVs isolated from early passage MSCs had a smaller median diameter (74.0 nm) than those isolated from the late passage did (98.8 nm). In Appendix A, in the bottom panel, the EVs isolated via SEC (85.4 nm) and those separated via UC (84.0 nm) were not different in size. This is likely due to the observer’s selection of EV particles for measurement via TEM. In contrast, both the DLS and NTA measures automatically counted all particles. Thus, the size density distribution influences the DLS and NTA measurements, not TEM size measurements.

### 2.4. EV Characterization via Dynamic Light Scattering (DLS)—Polydispersity Index (PDI) and Surface Charge (Zeta Potential)

DLS is used to measure the polydispersity index (PDI) and zeta surface potential, as shown in Appendix A. No differences in PDI were noted between the passages (Appendix A). There was a trend for EVs from late passage cells to have a higher PDI than that of early passage cells (0.48 vs. 0.46, respectively). In contrast, as shown in Appendix A, the PDI differed between EVs isolated via UC and SEC: EVs separated via UC had a PDI median value of 0.38, indicating a more monodispersed sample compared to the PDI of 0.55 for SEC samples. To take a closer look at the effect of the isolation method on the PDI, the PDI was plotted vs. CPD in Appendix A. This figure shows that PDI measured in UC EV samples was less dispersed and more strongly influenced by CPD than the PDI measured in SEC EV samples. Since the PDI is directly related to DLS’ size estimated, the lower PDI in UC EV samples may be caused by the smaller hydrodynamic size observed via DLS in the UC samples.

Appendix A shows the EV zeta potential results. Using ANOVA, we found the significant main effects of cell passage, isolation method, and interactions (passage x isolation) on the zeta potential of EVs. The main effects (passage) and (isolation method) are shown in Appendix A, respectively. Early passage EVs had a more negative surface charge than later passage EVs did. SEC isolation produced EVs with a lower (more negative) zeta potential than UC isolation did (Appendix A). The interaction term is shown in Appendix A; the zeta potential did not change in SEC-isolated EV samples when given more time in the culture. In contrast, EVs isolated via UC increased their zeta potential (became less negative), with increasing expansion, as indicated by the slope of the trend line.

### 2.5. Protein Content of EV Samples

The protein content results are shown in Figure 1. The EV samples isolated via UC had significantly different protein concentrations than the SEC samples did; about 10× higher protein concentration was found in the UC-isolated EV samples than that in the SEC-isolated samples. The median protein concentration was 38 µg/mL of UC-isolated samples, and a wide range of values (15–52 µg/mL) was found. EV samples isolated via SEC had a median protein concentration of 3 µg/mL and displayed a narrower range of values (1–8 µg/mL). The relationship between EVs and soluble protein was also evaluated using particle number per µg of protein (Figure 1G–I). Isolation via SEC yielded a significantly different number of (e.g., more) particles per µg of protein than UC did (~10×). SEC isolation yielded a median of 2.9 × 10^9^ particles/µg protein, which was compared to 3.3 × 10^8^ particles/µg protein obtained via UC isolation (8.8 × more particles per µg protein in the SEC isolation method). Obtaining more EVs per ug protein suggests that the SEC method produced a more “pure” EV sample, i.e., one with less protein contamination than that of UC (Figure 1G–I).

### 2.6. Dot Blot

EVs are characterized by proteins CD9, CD63, CD81, and ALIX [56], as shown in Figure 2. SEC- and UC-isolated EV samples were stained for CD9, CD63, CD81, ALIX protein, and tissue factor (TF). The protein blots demonstrated a trend in tetraspanins and TF staining associated with passaging (see Figure 2). Two masked reviewers scored the protein blot staining results, and the CD63 staining score was statistically different between early and late passages and was not different for the isolation method. At P2, 100% of EVs displayed the expression of CD63. At P5, only 33% of EVs isolates revealed CD63 expression at P12, no EV samples displayed CD63 expression. 

### 2.7. EV Procoagulant Activity Assay

The dot blot in Figure 2 shows the canine MSC EVs stained for TF. In Figure 3A, TF immunofluorescence of canine and human MSCs stained for TF. Only human MSCs were stained with anti-CD142 (functional clone HTF-1).

As shown in Figure 3B, canine EVs had mean procoagulant activity levels of 72.1 ± 6.1 ng/mL, which were not significantly different from those of human EVs (79.8 ± 7.9 ng/mL) or human MSCs (positive control, 80.1 ± 13.7 ng/mL). This finding indicates that canine EVs, human EVs, and human MSCs have similar procoagulant activity levels. FXa generation was measured for early passage (P2) and late passage (P12) EVs produced using both isolation methods to assess the procoagulant activity of canine MSC-EVs. As shown in Figure 4C, early passage EVs generated an average of 41.9 ± 34.3 ng/mL FXa, while late passage EVs generated 66.2 ± 43.2 ng/mL of FXa. As shown in Figure 3D, no difference was observed between the isolation methods. EVs isolated via SEC generated an average of 39.4 ± 39.6 ng/mL FXa, which was compared to UC-isolated EVs at 68.7 ± 36.1 ng/mL. Regardless of the passage or isolation method, EVs demonstrated similar procoagulant activity levels via the generation of TF-specific FXa. Taken together, these data indicate that canine MSC EVs are procoagulant due to the expression of TF. This is not influenced by passaging (i.e., time in culture) or the isolation method.

Procoagulant activity was calculated as FXa generated (in nM) per million EVs and per million cells, as shown in Table 1. The EV samples yielded a range of 0.002 to 0.048 nM FXa per 1 × 106 EVs. The average was 0.023 ± 0.017 nM FXa per 1 × 10^6^ EVs. When it was normalized to the cells producing EVs, the range was 41 to 1929 nM FXa per 1 × 10^6^ cells. The average was 519 ± 543 nM FXa per 1 × 10^6^ cells.

To confirm that the procoagulant activity demonstrated by EVs was due to the expression of TF, EVs were incubated with an anti-TF (blocking) antibody, before incubation with FVIIa and FX. As shown in Figure 4E, when canine EVs were incubated with an anti-TF antibody (polyclonal), FXa generation was not inhibited (average ± SD: 63.4 ± 20.2 vs. 75.9 ± 11.4 ng/mL, for canine EVs plus polyclonal antibody). Similarly, as shown in Figure 4F, when human EVs were incubated with the anti-TF antibody (polyclonal), FXa generation was not inhibited (average ± SD: 35.8 ± 7.2 vs. 27.0 ± 4.4 ng/mL, for human EVs plus polyclonal antibody; right bar in Figure 4F). In contrast, when human EVs were incubated with a TF antibody described as a functional inhibitor (clone HTF-1), e.g., a clone previously shown to not be reactive with canine TF and to inhibit FXa generation, FXa generation was inhibited (average ± SD human EVs plus HTF-1 antibody: −6.1 ± 24.3 ng/mL; second to right bar in Figure 3F).

## 3. Discussion

Here, we hypothesized that MSC-EVs would express TF such as MSCs do. To address this hypothesis, EVs were derived from canine umbilical cord-derived MSCs using two different isolation methods and characterized as suggested by the International Society of Extracellular Vesicles (ISEV) recommendations. We found that canine MSCs, canine MSC-EVs, and human MSC-EVs all expressed tissue factor and displayed procoagulant activity. These findings suggest that hemocompatibility is a consideration to factor into the clinical equation of MSCs and MSC-EVs.

### 3.1. EV Characterization

ISEV recommends EV characterization using several methods, for example, DLS, NTA, TEM, and protein surface markers [56] (note that details about these can be found in the Appendix A). SEC and UC produced EVs size estimates in the range associated with exosomes [7,10,27,60,61,62]. EVs isolated via SEC were 12% larger than those isolated using UC when analyzed by NTA, and 26% larger when analyzed by DLS, and there was no difference in size when measured by TEM. This suggests that “contamination” by small species of particles in UC affected the size measurements in NTA and DLS, but not TEM. When the relationship between EV size and culture time (either passage or CPD) was evaluated, the EV size remained unchanged when measured via DLS or NTA, but they increased in size during late passaging when they were measured via TEM. The former observation suggests an absence of the accumulation of apoptotic bodies or other larger vesicles that might be associated with cell senescence [63,64]. The EVs isolated here had EV-characteristic protein surface markers: CD9, CD63, CD81, and Alix (short for Apoptosis-Linked gene-2-Interacting protein X). The appearance of EVs in TEM had the EV-characteristic doughnut shape. Taken together, our data support the notion that exosomes were the type of EVs isolated via both SEC and UC.

### 3.2. Assaying Hemocompatibility

The procoagulant activity was assayed in two ways. First, since the authors of only a few previous works examined canine TF expression, TF staining was examined, and both canine MSCs and MSC-EV expressed TF (Figure 4A). As reported here, Gruber et al. reported that canine mammary cancer cells highly express TF [65]. Second, procoagulant activity was assessed via a functional assay that measured TF-specific FXa generation. Here, we ran into an “issue”, namely there are no commercial sources of canine FVIIa and FX. The alternative options included human, rodent, and bovine ones, and since Knudsen et al. reported that human FVIIa binds to canine TF similarly to human ones [55], we chose to use human ones. The assay used here was modified from previously reported EV assays, and we measured the procoagulant activity of MSC-EVs [49,55]. However, this assay varied greatly despite performing technical triplicates and using six different MSC lines in two passages (P2 and P12). This could be due to biological variability (i.e., line-to-line variation), the sample size, or other sources of experimental variation.

This is the first report of canine MSC-EV procoagulant activity, and the effect size observations should assist with the design of future work. As noted by others, this assay is time-consuming, labor-intensive, expensive, and has high inter-assay variability [49,55]. Although the reactions were performed with a consistent number of EVs, high variability in particle count estimates provided by the NTA could be a factor here. These limitations make this assay unsuitable for screening EVs for clinical use as a safety measure. The clinical translation of this assay requires optimization to decrease the variability, time, amount of labor, and cost.

We compared FXa generation using human MSCs, human MSC-EVs, and canine MSC-EVs and found similar procoagulant activity levels across groups. This observation confirms the cross-species compatibility and feasibility of this modified assay for MSC-EVs. Next, since previous reports speculated that the TF expression of MSCs may be a product of cell culture [16], we compared TF-specific FXa generation in early (P2–P3) and late (P11–P12) passages. We used EVs isolated via SEC and UC. Our study also found similar procoagulant activity levels in early and late passage EVs, with a trend for increased activity in late passage EVs (66.2 ng/mL vs. 41.9 ng/mL for early passages). The isolation method also showed more coagulation activity in UC-isolated EVs than that in SEC-isolated EVs (68.7 ng/mL vs. 39.4 ng/mL, respectively). Thus, our findings differed from those of Oeller et al., who reported increased TF expression with MSCs during passaging [16].

Hisada et al. reported TF-specific procoagulant activity in EVs from human plasma samples, with four response categories ranging from zero (<0.5 pg/mL) to strong (>2 pg/mL) [45]. MSC-EVs fell within the strong category based on their classification for EVs from platelet-free plasma, with an average procoagulant activity of 54,040 pg/mL. The much larger value exhibited by MSC-EVs compared to that of the EVs from platelet-free plasma could be due to differences in the assay or specific EVs’ activity. This comparison is concerning because MSC-EVs display higher TF activity levels than what is considered to be “strong” by others. In contrast, Che et al. reported TF-specific activity in EVs from the breast cancer cell line, MDA-MB-231, by normalizing FXa generation per million tumor cells [49]. Breast tumor cell EV activity was observed in 1 and 2 nM per million cells [49]. When our MSC-EV’s activity levels were normalized per million MSCs, the procoagulant activity level was lower, ranging between 0.026 and 0.854 (average 0.329 ± 0.28) nM per 1 × 106 MSCs. Thus, the procoagulant activity level of MSC-EVs is lower than it is in breast cancer cells, but higher than it is in platelet-free plasma.

A drawback of the two EV procoagulant assays previously discussed is that a set volume of EVs was used per reaction well. This creates difficulties when values across laboratories are compared because the number of EVs generating FXa varies. Here, we used a set number of EVs (5 × 10^7^) per reaction well and an equal volume of buffer. This enabled consistency in the assays, so that the values exhibited are caused by differences in the cell line, passage, or isolation and not the number of EVs.

Here, we report the procoagulant activity (FXa generation) per million of EVs for the first time. The samples ranged from 0.002 to 0.045 (0.023 ± 0.02) nM FXa per million EVs. This value could be used to compare EV studies, biological starting material, parental cell type, and other factors. This value also provides an estimate of procoagulant activity that can be calculated for a “dose” of EVs used in a clinical setting.

When the data were normalized per million MSCs or per million EVs, two-thirds of the lines exhibited more FXa generation by EVs in the late passages compared to those in the early passages. Thus, one-third of the lines showed more FXa generation by EVs from the early passages compared to those of the late passages. We have no explanation for this result, other than “biological variability”. These data indicate that a pattern may exist, but a larger sample size would be needed to resolve it. Another possibility is that other physiological characteristics, such as the sex of the donor, breed of dog, etc., may impact these observations. Unfortunately, the studies needed to resolve these issues are beyond the scope of this initial report.

### 3.3. Limitations of the Present Work

We attempted to confirm that the specificity of procoagulant activity exhibited here by canine MSC-EVs was due to surface TF expression caused by incubating them with a TF “functional” antibody. The TF antibody clone HTF-1 inhibits human TF activity by blocking FXa generation [49,66,67,68]. The HTF-1 clone was not functional at inhibiting the canine TF epitope. Next, we tried to inhibit FXa generation using the canine-reactive polyclonal antibody (the clone used in dot blots). Still, that antibody could not inhibit FXa generation by canine EVs either. Thus, we demonstrated that human MSCs and their EVs exhibited similar levels of procoagulant activity to that of canines and that FXa generated by human MSCs and MSC-EVs was inhibited by the HTF-1 clone. The findings suggest that FXa generation is due to TF expression by canine MSC-EVs, which is similar to what we observed in human MSC-EVs. Still, we could not prove specificity by blocking FXa generation with an antibody.

Here, TF expression and FXa generation were demonstrated in vitro. The quantitation of TF expression using a TF standard protein curve was unsuccessful and required troubleshooting and optimization or the use of another quantification method, such as a Western blot. Quantification via Western blot was not pursued here due to the low protein content of the SEC-isolated samples. Therefore, a stronger case for patient risk could be generated by performing a quantitative assay, such as a fibrin generation assay or an intravascular thrombosis induction assay [16,52].

### 3.4. Impact of Our Findings on Clinical Translation

MSCs have been investigated as a potential therapeutic agent in many clinical modalities because they stimulate tissue regeneration and create a variable, localized anti-inflammatory effect. MSCs are thought to exert therapeutic effects mainly through the production and secretion of soluble factors and EVs. EVs play a role in intercellular communication and signaling, antigen presentation, cell adhesion, inflammation, and tissue remodeling. In addition, EVs share traits with their parental cell types, such as protein expression and cargo. These physiological properties suggest that MSC-EVs represent a potential cell-free therapeutic agent. Here, the effects of MSC expansion and EV isolation methods were examined in canine MSC-EVs. The results indicate that it is important to consider both parameters EV manufacturing scale-up and clinical translation. Furthermore, MSC-EVs express TF and procoagulant activity, such as their parental cells do [11,13,16,22]. Thus, thromboembolism is a risk and safety concern for the clinical application of allogeneic MSCs, mainly because MSCs become trapped in the lungs and other organs after an intravenous injection. The risk is concerning since MSCs are being explored as a therapeutic agent to treat respiratory complications of coronavirus-induced disease (COVID-19) [4,69,70,71]. Thus, the procoagulant activity of MSCs and their EVs may serve as a screening tool in specific clinical settings [16].

In conclusion, MSC-EVs, particularly exosomes, may offer the same therapeutic benefits as MSCs do, without the risks and complications associated with the cellular product [67,68]. The fact that EVs share many traits with their parental cells potentially makes them a double-edged sword. On the one hand, there is the potential for EVs to share the therapeutic benefits of MSCs. On the other hand, as demonstrated here, MSC-EVs share TF expression and procoagulant activity with MSCs. It is reasonable to assume that MSC-EVs share a potential to induce thrombosis and the formation of micro-emboli with MSCs. Given that the procoagulant activity of human MSCs and tumor-derived EVs is well known, the assumption that MSC-EVs also possess procoagulant activity is sensible [44,46,48,49,50,51,53,72], and steps to mitigate this risk should be considered in clinical settings.

## 4. Materials and Methods

### 4.1. Preparation of Conditioned Media from Canine Umbilical Cord-Derived Mesenchymal Stromal Cells

Six canine umbilical cord-derived MSC (CUC-MSC) lines previously characterized [58] that had demonstrated the ability to expand over fifteen passages were used in this experiment (Figure 4A and Appendix A). Briefly, cryopreserved CUC-MSCs (P1) were thawed and seeded at a density of 2 × 104 cells/cm^2^ on gelatin-coated T150 tissue culture vessels. MSC lines expanded faster with each passage from P2–7 (Appendix A) or 0 during about 15 cumulative population doublings (CPD, see Appendix A). Generally, MSC lines showed similar expansion characteristics; CUC line 20 and CUC line 30 represent the two extremes (see Appendix A).

The expansion medium (Dulbecco’s Modified Eagles Medium (DMEM, Gibco, New York, NY, USA, Catalog No. 11965092) supplemented with 10% fetal bovine serum (HyClone, GE Healthcare Life Sciences, Freiburg, Germany, Catalog No. SH3007103), 1% antibiotic-antimycotic (Gibco, Catalog No. 14190250; Gibco, Catalog No. 15240062), 1% Glutamax (Gibco, Catalog No. 35050061), and ten ng/mL fibroblast growth factor-basic (bFGF, Gibco, Catalog No. PHG0264) was replaced with DMEM (0.33 mL/cm^2^) without fetal bovine serum (FBS) or other supplements when flasks reached 60–70% cell confluence. After 24 h, CM was collected and clarified to remove cells and cellular debris as described [57]. Before EV isolation, CM was stored at −80 °C since previous work found this to be the optimal storage condition for MSC-derived EVs [57]. EV consistency between cell lines was assured using cell lines with viability ≥ 95% during passaging.

Human umbilical cord mesenchymal stromal cells were isolated, expanded, and characterized as described in [57,73,74].

### 4.2. Design of Experiments Approach

As shown in Figure 4B, the strategy implemented used an 8 × 2 factorial design, with factor A being passaged at eight levels (P2, P3, P4, P5, P9, P10, P11, and P12) and factor B using the EV isolation method with two groups: size exclusion chromatography, SEC; ultracentrifugation, UC. The experimental design was balanced to ensure that equal observations were made of all combinations to minimize the impact of cell line-to-cell line variation on the results. Three of the six cell lines were assigned randomly to each isolation method. Passages 2–5 were grouped and considered to be “early passages”. Passages 9–12 were grouped and considered to be “late passages”. Middle passages (P6–P8) were collected, stored, and are not reported here. The population doubling time (PDT) was calculated for each cell line during each passage. An estimate of the number of adherent cells was calculated using the PDT and the number of hours the cells were in culture before the removal of serum-containing media. Note that cell culture expansion data can be found in the Appendix A.

### 4.3. EV Isolation via a Combination of Ultrafiltration and Size-Exclusion Chromatography (SEC)

Briefly, EVs were isolated from CM using a combination of ultrafiltration and size-exclusion chromatography using our published procedures [57]. EV-containing fractions were pooled, divided into 1 mL aliquots in polypropylene microcentrifuge tubes, and stored at −80 °C until characterization.

### 4.4. EV Isolation via Ultracentrifugation (UC)

Briefly, EVs were isolated from CM via ultracentrifugation with slight modifications [26] (see Appendix A for details). The resulting pellet was suspended in 6 mL of Dulbecco’s phosphate-buffered saline (DPBS), vortexed, divided into 1 mL aliquots, and stored at −80 °C until characterization.

### 4.5. Lyophilization of EVs

To freeze dry the aliquots, they were removed from −80 °C and loaded into a TF-10A 1.2-L vacuum freeze dryer (Tefic Biotech Co., Limited, Xi’an, China) [26,57]. Samples were lyophilized in batches of six for 16–18 h following the manufacturer’s protocol. Following lyophilization, samples were stored at room temperature and sealed with parafilm to protect them from moisture absorption. Immediately before use, aliquots were rehydrated with sterile UltraPure distilled water (Invitrogen, Waltham, MA, USA, Cat. No., 10977) to 10% of their original volume, and then briefly vortexed. The protein concentration of the reconstituted sample was measured using absorbance at 280 nm with a spectrophotometer. The reconstituted samples were analyzed via transmission electron microscopy (TEM) and dot blots for EV marker proteins.

### 4.6. Nanoparticle Tracking Analysis (NTA)

NTA was used to estimate the EV size distribution and concentration using a NanoSight LM-10 (Malvern Pananalytical Ltd., Malvern, UK) using settings that have been previously described [57,75] (see the Appendix A for details). The particle counts per ml (i.e., concentration) was derived from the average of 5 technical replicates for comparison via two-way repeated measures analysis of variance (ANOVA), with the factors being the passages (early (P2–P5); late (P9–P12)) and isolation method (UC and SEC). Following the observation of significant ANOVA main effects or interactions, planned pairwise comparisons were made using the Holm–Sidak method. Cell passages were grouped into early (P2–P5) or late (P9–P12) for comparison via the Mann–Whitney rank sum test with Yates continuity. The size measurement was made from the five technical replicates and averaged for comparison using the Mann–Whitney rank sum test.

Using NTA concentration data, the number of EVs released per cell was calculated as previously described [57]. The assumption was that cell proliferation ceased when cell culture media were removed. The number of EVs released per cell was estimated as the total particles per ml in the sample divided by the estimate of cells in culture at the time of media removal. The number of EVs released per cell were averaged by passage and compared using repeated measures ANOVA on ranks. To assess correlation, a simple linear regression analysis was performed, with particles per ml per cell as the dependent variable vs. cumulative population doublings (CPD). A second round of linear regression analysis was performed, with particles per cell as the dependent variable vs. population doubling time (PDT). Linear regression was performed to compare the results obtained from each isolation method.

### 4.7. Dynamic Light Scattering (DLS)

DLS was used to analyze the hydrodynamic size distribution, surface charge properties, membrane integrity, and overall EV stability as previously described [26,57]. See the Appendix A for DLS data.

### 4.8. Transmission Electron Microscopy (TEM)

TEM was performed to visualize EV morphology and estimate the EV size as previously described [26,57]. See the Appendix A for details of TEM characterization and TEM data.

### 4.9. Protein

The protein concentration of samples isolated via UC and via SEC was estimated using a Pierce BCA protein assay kit (Thermo Scientific, Waltham, MA, USA, Cat. No. 23225) according to the manufacturer’s instructions, as previously described [26,57]. See the Appendix A for expanded details about protein characterization and protein characterization data.

As an estimate of EV sample purity, EV particle concentration per µg of protein was calculated as previously described [57,59]. The particle count per mL of the sample determined via NTA was divided by the protein concentration (µg/mL) determined via the BCA assay to give an isolated particle concentration per µg of protein. Similarly, the changes in protein during passaging, PDT, or CPD were evaluated via linear regression. Measurements obtained from six MSC lines in eight passages were assessed by isolation and compared using the Mann–Whitney rank sum test with Yates continuity correction. See the Appendix A for details and the dataset.

### 4.10. Immunocytochemistry

To detect cell membrane expression of tissue factor (CD142), human and canine UC-MSCs were grown in cultures until the plates were 80–90% confluent [26,58,73,74]. Once this confluency was obtained, the medium was removed, and the MSCs were washed twice with sterile DPBS containing calcium and magnesium (Gibco, Cat. No., 14040-133), and then fixed with freshly depolymerized 4% paraformaldehyde in 10 mM phosphate buffer (pH 7.4) for 30 min at room temperature. The fixed cells were washed three times with DPBS before staining. Nonspecific binding was blocked with DPBS supplemented with 0.2% Triton X-100 (Sigma Aldrich, St. Louis, MO, USA, Cat. No., X100), 0.2% gelatin, 1% horse serum, and 1% goat serum. MSCs were incubated with a primary antibody to CD142 (Table 1, 1:200 dilution) overnight at 4 °C. The following morning, cells were washed three times with DPBS and stained with secondary antibody goat anti-rabbit IgG Alexa Fluor 488 (1:200 dilution, Life Technologies, Carlsbad, CA, USA, Cat. No., A-11008) for three hours at four °C and protected from the light. Cells were washed three times, and DNA was stained with 4′,6-diamidino-2-phenylindole (DAPI, 1 µg/mL, Sigma Aldrich, Cat. No., D9542) for 15 min at room temperature and rinsed with DPBS. Canine UC-MSCs not exposed to the primary antibody and processed throughout all other staining steps served as negative controls. Human UC-MSCs were stained as described above and served as positive controls. Fluorescent and phase contrast images were captured with an EVOS FL Auto imaging system (Life Technologies).

### 4.11. Dot Blot

Lyophilized EVs were rehydrated using sterile water and blotted onto a PVDF membrane to stain EV proteins. Forty-eight blots were prepared; the original blots can be found in the Appendix A. Membranes were stained to detect CD9, CD63, CD81, ALIX, and TF using antibodies listed in the Appendix A. Each stained blot was scored by three investigators blinded to the experimental conditions; a positive result was given a score of 1, and a negative expression was assigned a score of 0. Total scores for individual markers were analyzed using the Mann–Whitney rank sum test. Scores are given in the Appendix A spreadsheet.

### 4.12. Procoagulant Assay

The procoagulant activity of EVs isolated from six UC-MSCs was assessed in the early (P2–P5) vs. late (P9–P12) passage cells using a protocol proposed by Che et al., with modifications [49]; the Appendix A provide the details.

To calculate procoagulant activity, the amount of activated factor X (*FXa*) generated (determined from the standard curve) was divided by the reaction volume in the well according to the formula:$$Procoagulant\;FXa\;Activity=\frac{FXa\;generated\;\left(\mathrm{ng}\right)}{Volume\;in\;each\;well\;\left(\mathrm{mL}\right)}$$

From the procoagulant activity, the concentration of *FXa* (nM) was reported by dividing the procoagulant *FXa activity* (formula above) by its molecular weight (46 kDa). *FXa* generation per 1 × 10^6^ EVs was calculated as the *FXa* generation (in nM) divided by millions of EVs in the reaction well.

### 4.13. Statistics

After checking the ANOVA assumptions, ANOVA was used to evaluate main effects (and interactions). After finding a significant ANOVA main effect or interaction, the post hoc testing of pre-planned comparisons was conducted using either the Bonferroni’s or Holm–Sidak correction. Data are presented as boxplots. The box gives the median, first, and third quartiles; the whiskers show the 10th and 90th percentiles. Any potential outliers are shown as open circles. Sometimes, data are presented as mean (average) ± one standard deviation. If ANOVA assumptions were not met, Kruskal–Wallis ANOVA on ranks was used. For pairwise comparisons, statistical assumptions were confirmed, and data were analyzed using Student’s t-test. In all cases, the hypothesis testing was two-tailed, and a *p* < 0.05 was considered to be “significant”. A power analysis determined the sample size needed to detect significant differences at a desired power of 0.8 and an alpha of 0.05. Linear regression was conducted to identify trends. The entire dataset was used for statistical testing and is provided in the Appendix A.

### 4.14. Data Visualization

Statistics and graphing were performed using SigmaPlot (version 14.5, build 14.5.0.101, Systat Software, Inc., San Jose, CA, USA). Graphs were saved as 600 dpi *.EMF files. Labeling or arranging them into figures was achieved using Canvas X (version 19, build 333, Canvas GFX, Inc., Boston MA, USA), and those files were saved in Canvas’ native file format (e.g., *.CVX). Canvas was used to save the figures at 300 dpi with indexed color as uncompressed *.TIFF files.

## Figures and Tables

**Figure 1 ijms-24-09216-f001:**
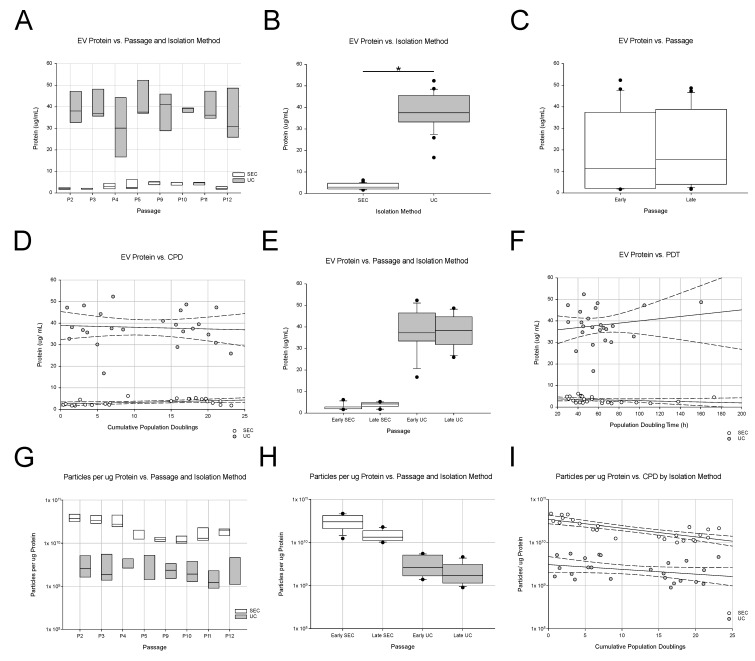
Protein concentration of EV samples. (**A**–**C**) The effect of passage on protein content was not obvious. However, a frank difference was noted between the protein contents of size-exclusion chromatography-isolated EVs (SEC, white bars) and ultracentrifugation-isolated EVs (UC, gray bars). (**D**,**E**) When protein content was plotted vs. cumulative population doublings (CPD), the trend lines suggested that protein content in EVs tended to increase with longer passage times. This trend was more apparent in SEC-isolated EVs. (**F**) When we studied whether protein content was affected by population doubling time, no big impact of population doubling time was noted. (**G**–**I**) When the particles per ug protein were calculated, UC and SEC isolated EVs had fewer particles per ug protein in late passages. The asterisk indicates *p* < 0.05.

**Figure 2 ijms-24-09216-f002:**
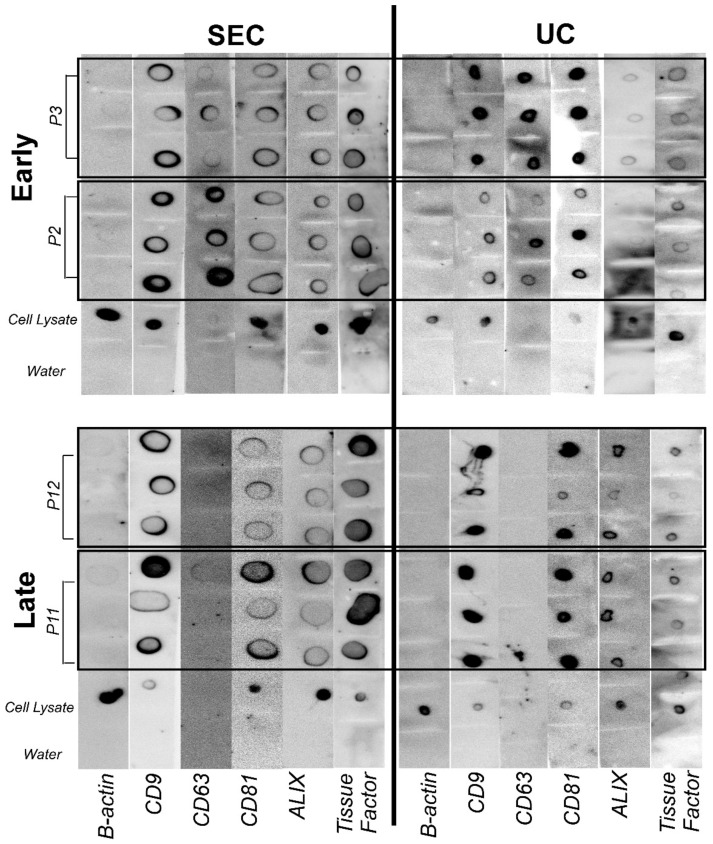
Extracellular vesicles (EVs) isolated from canine umbilical cord-derived mesenchymal stromal cells (MSCs) express tissue factor (TF) over passages but differ in terms of the expression of CD63 between early and late passages. Both early (P2 and P3 shown) and late passage (P11 and P12 shown) EVs expressed TF regardless of the EV isolation method (size-exclusion chromatography, SEC, or ultracentrifugation, UC). The expression of clusters of differentiation (CD) 9, CD63, CD81; ALIX; TF; protein loading control, ß-actin, via negative control (water), whole-MSC cell lysate (positive control) in both early (top) and late (bottom) passages, as well as EVs isolated via SEC (left) and UC (right). EVs express CD9, CD81, and ALIX regardless of passaging or isolation. However, irrespective of the isolation method, CD63 expression was not observed in late passage EVs.

**Figure 3 ijms-24-09216-f003:**
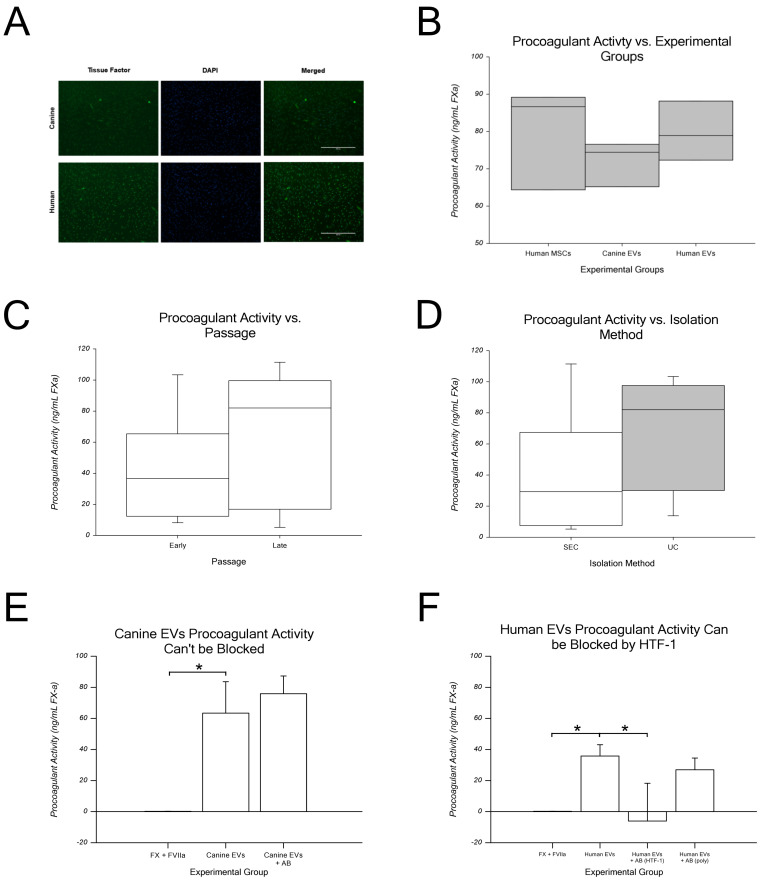
Procoagulant activity of MSCs and EVs. (**A**) Tissue factor (TF) was expressed by human and canine umbilical cord-derived mesenchymal stromal cells. Note that the staining intensity of human MSCs (bottom) appears to be higher than that of canine MSCs. Calibration bar = 400 µM. (**B**) The procoagulant activity (FXa generation) of human and canine extracellular vesicles (EVs) isolated from umbilical cord mesenchymal stromal cells (MSCs). No significant differences in procoagulant activity levels were found between human MSCs (positive control, left bar) and canine (middle) or human EVs. Data from three human MSC lines from passages 4 to 6 and three human and canine EV samples. (**C**) Procoagulant activity (ng/mL of FXa generated) of EVs isolated from canine umbilical cord MSCs via cell passaging: early (P2–P5) vs. late (P9–P12) passage. No statistical differences were found, but the trend was for late-passage EVs to have higher levels of procoagulant activity. (**D**) No significant differences were found between the isolation method used: size exclusion chromatography- (SEC, white bars) vs. ultracentrifugation (UC, gray bars)-based EV isolation. There was a trend for EVs isolated via UC to have a higher procoagulant activity level than that of SEC. Data from EVs were derived from six canine MSC lines; there were three in each isolation method. (**E**) Canine EVs had significantly higher procoagulant activity levels than the negative control did (FX + FVIIa). However, when they were incubated with polyclonal anti-tissue factor antibody, FX-a generation was not inhibited in K9 EVs. (**F**) Human EVs displayed significantly more procoagulant activity than the negative control did (FX + FVIIa). The procoagulant activity of human EVs was significantly inhibited by an anti-TF antibody (clone HTF-1 was previously shown to inhibit the function of TF). In contrast, and similar to canine EVs, procoagulant activity was not inhibited by the polyclonal TF antibody (far right). Data are presented as the average of three biological replicates ± standard deviation. The asterisk indicates *p* < 0.05.

**Figure 4 ijms-24-09216-f004:**
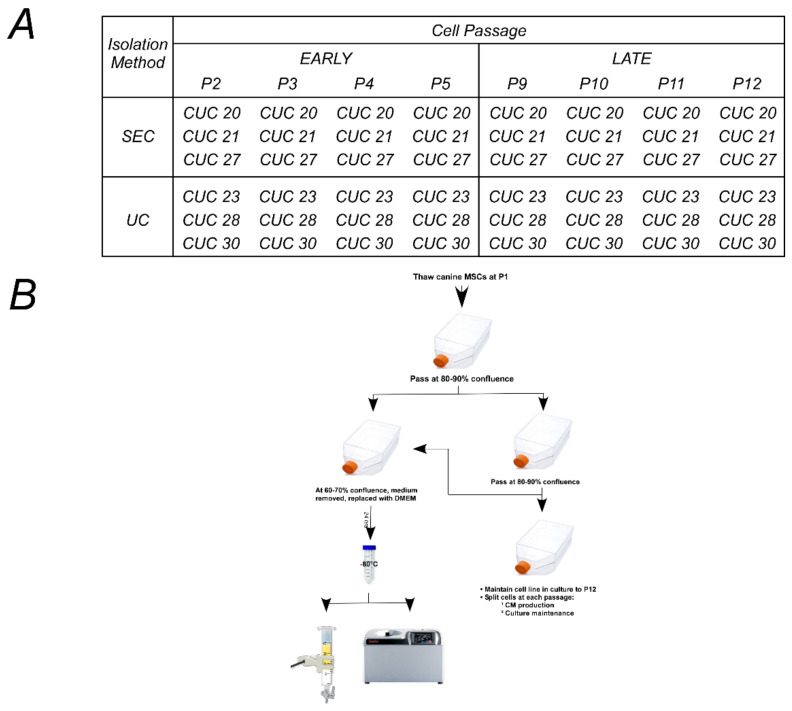
Experimental design and schematic. A randomized, 8 *×* 2 factorial is shown. (**A**). Design of experiments protocol parsed into two factors. Factor A (Cell Passage) had eight levels: P2, P3, P4, P5, P9, P10, P11, and P12. Factor B (EV isolation method) had two levels: ultracentrifugation (UC) and size-exclusion chromatography (SEC). Six canine umbilical cord-derived mesenchymal stromal cell lines (CUC) selected from a cell bank were used and randomly assigned to levels of Factor B. (**B**). Experimental schematic: Canine umbilical cord-derived mesenchymal stromal cells (MSCs) were thawed at passage 1 (P1), plated at a density on gelatin-coated T-150 flasks, and cultured as previously described [58]. Once plates reached 80–90% confluence, MSCs were passaged and plated into two T-150 flasks. Flask (1) was used for the production of conditioned medium (CM) (arrow going to the left side): cell culture medium of MSCs at 60–70% confluence was removed and replaced with DMEM for 24 h. This medium was considered to be “conditioned” and was collected in a 50 mL centrifuge tube. The CM was stored at −80 °C until isolation via size-exclusion chromatography (SEC) or ultracentrifugation (UC) [59]. Flask (2) was used to maintain the MSC line in culture (arrow pointing to the right side). In Factor A, passages 2–5 were defined as “early passage”, and passages 9–12 were described as “late passage”.

**Table 1 ijms-24-09216-t001:** Procoagulant activity (FXa generated) per million cells and EVs.

Line	Cell Passage	Isolation Method	EV Number (per well)	FX-a Activity (ng/mL)	FX-a Generated (nM)	FX-a Generated per 1 × 10^6^ EVs (nM)	FX-a Generated per 1 × 10^6^ Cells
CUC 20	Early	SEC	5.00 × 10^7^	37.7	0.82	0.016	260.114
CUC 20	Late	SEC	5.00 × 10^7^	111.4	2.42	0.048	533.752
CUC 21	Early	SEC	5.00 × 10^7^	52.8	1.15	0.023	735.681
CUC 21	Late	SEC	5.00 × 10^7^	5.2	0.11	0.002	40.688
CUC 23	Early	UC	5.00 × 10^7^	103.4	2.25	0.045	1928.998
CUC 23	Late	UC	5.00 × 10^7^	95.7	2.08	0.042	1109.115
CUC 27	Early	SEC	5.00 × 10^7^	8.2	0.18	0.004	59.294
CUC 27	Late	SEC	5.00 × 10^7^	20.8	0.45	0.009	154.836
CUC 28	Early	UC	5.00 × 10^7^	35.4	0.77	0.015	198.870
CUC 28	Late	UC	5.00 × 10^7^	89.9	1.95	0.039	405.535
CUC 30	Early	UC	5.00 × 10^7^	13.8	0.30	0.006	202.521
CUC 30	Late	UC	5.00 × 10^7^	74.2	1.61	0.032	597.879
					Mean ± SD	0.023 ± 0.017	518.940 ± 543.138

## Data Availability

The original datasets generated here are available after completing the Material Transfer Agreement.

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
