# Peer review of "Procoagulant Activity of Umbilical Cord-Derived Mesenchymal Stromal Cells’ Extracellular Vesicles (MSC-EVs)"

_ijms, 2023, doi:10.3390/ijms24119216_

Round 1
Reviewer 1 Report
The manuscript by Wright and colleagues possesses scientific novelty and explores the safety of EV-MSC. The work could be of interest for specialists in cell therapy and MSC/EV biology.
some remarks:
Why study both human and canine EV-MSC? Authors could outline the reason.
What is the reason to use dot-blot instead of western-blot for EV analysis?
Discussion should be restructured. The title of the paper is "procoagulant activity...", but most of the discussion is devoted to the details of EV isolation and characterisation. Some of the text should be moved to the "results" section. DoE is not mentioned neither in methods nor in results.
The quality of Fig 4a is very low. Hard to make any conclusions based on the presented microscopy.
AD-MSC is more typical word than AT-MSC
Author Response
Reviewer #1:
The manuscript by Wright and colleagues possesses scientific novelty and explores the safety of EV-MSC. The work could be of interest for specialists in cell therapy and MSC/EV biology.
some remarks:
Why study both human and canine EV-MSC? Authors could outline the reason.
RESPONSE TO REVIEWER 1: Thank you for your question. We addressed your query by adding a sentence to the introduction. We should have explained why we used EVs from dogs rather than humans for this study.
>> ADDED TO MANUSCRIPT on line 81: “…Since the dog is considered an excellent large animal model for human inherited bleeding disorders and contributed to the understanding of human tissue factor and Factor VIIa complex [1,2], here, we evaluated canine MSC-derived EVs for TF-expression and procoagulant activity. “
Two references were added.
- What is the reason to use dot-blot instead of western-blot for EV analysis?
RESPONSE TO REVIEWER 1: Thank you for your question. Both dot and western blot can characterize protein by visualizing antibody binding to the sample. Western blot can convey size information that dot blot does not. In our first EV paper, where we used ultracentrifugation (UC) to isolate EVs [3], we had inconsistent western results, perhaps due to poor transfer from the gel to the membrane or the transferred proteins did not stick well. When we started using size exclusion chromatography (SEC) to isolate EVs, the sample’s protein content was much lower than UC, as shown in the present report. We found that we could obtain consistent dot blot results by freeze-drying the sample and reconstituting it at 10x higher concentration. We provided a dot blot to characterize the EV per the International Society of Extracellular Vesicle’s publication standards for EV [4].
>> NOTE: No changes to the manuscript.
Discussion should be restructured. The title of the paper is "procoagulant activity...", but most of the discussion is devoted to the details of EV isolation and characterisation. Some of the text should be moved to the "results" section.
RESPONSE TO REVIEWER 1: Thank you for your suggestion. We agree that the Discussion section was too long and that we could improve the take-home message by removing redundancy and working on the flow. The Discussion was trimmed by eliminating 185 lines, and these lines were moved to the Supplemental file, Discussion section. In addition, the Discussion was rewritten to improve flow and focus. Again, thank you for pointing out this important and much-needed change.
>>> Changes to Ms:
- Lines 387-572 were moved to the Supplemental file, Discussion section.
- One New heading was added to the Supplemental file, Discussion section.
- The entire Discussion was rewritten to tighten and improve flow and focus.
- Section headings were added to the Discussion to improve focus and flow.
- DoE is not mentioned neither in methods nor in results.
RESPONSE TO REVIEWER 1: Thank you for carefully reading the manuscript. The second paragraph of the Methods was “Experimental Design”. We changed that heading to “Design of Experiments Approach” to clarify the design of experiments (DoE) approach used here. The DoE refers to the systematic, structured design, allowing us to efficiently examine multiple variables (i.e., isolation methods and culture expansion) and responses. The results reflect this approach.
>> Line 641 of the manuscript was changed from “Experimental Design” to “Design of Experiments Approach”.
The quality of Fig 4a is very low. Hard to make any conclusions based on the presented microscopy.
RESPONSE TO REVIEWER 1: Thank you for your input. We agree that seeing the surface staining using a 10x objective is difficult. We found another set of micrographs that depicted the surface staining of canine MSCs a little better and utilized these to remake Fig 4a, top panel.
>>> Replaced Figure 4a top panel to improve the quality of micrographs.
AD-MSC is more typical word than AT-MSC
RESPONSE TO REVIEWER 1: Thank you for catching our oversight. We corrected these errors in the manuscript per your suggestion.
>> Lines 42, 45, 46, 55 corrected: AT-MSC changed to AD-MSC
References
- Knudsen, T.; Olsen, O.H.; Petersen, L.C. Tissue factor and factor VIIa cross-species compatibility. Front Biosci (Landmark Ed) 2011, 16, 3196-3215, doi:10.2741/3906.
- Nichols, T.C.; Hough, C.; Agerso, H.; Ezban, M.; Lillicrap, D. Canine models of inherited bleeding disorders in the development of coagulation assays, novel protein replacement and gene therapies. J Thromb Haemost 2016, 14, 894-905, doi:10.1111/jth.13301.
- Abello, J.; Nguyen, T.D.T.; Marasini, R.; Aryal, S.; Weiss, M.L. Biodistribution of gadolinium- and near infrared-labeled human umbilical cord mesenchymal stromal cell-derived exosomes in tumor bearing mice. Theranostics 2019, 9, 2325-2345, doi:10.7150/thno.30030.
- Thery, C.; Witwer, K.W.; Aikawa, E.; Alcaraz, M.J.; Anderson, J.D.; Andriantsitohaina, R.; Antoniou, A.; Arab, T.; Archer, F.; Atkin-Smith, G.K.; et al. Minimal information for studies of extracellular vesicles 2018 (MISEV2018): a position statement of the International Society for Extracellular Vesicles and update of the MISEV2014 guidelines. J Extracell Vesicles 2018, 7, 1535750, doi:10.1080/20013078.2018.1535750.

Reviewer 2 Report
This study is interesting with clinical significance. MSC-EV is a popular candidate for cytotherapy. It is necessary to evaluate the therapeutic risk of MSC-EV.
The authors put forward a new point of view for the therapeutic risk of MSC-EV via procoagulant activity. The followings are some comments to the authors.
Comments:
1.I suggest that specific data on the results should be added in the Abstract.
2.If abbreviation had been defined in the text when used for the first time, abbreviation is recommended in the text below. For example, in line 62 "Extracellular vesicles (EVs) ".
3.Please define all abbreviations in the text when used for the first time. For example, line 100"UC”,and line 101"SEC".
4. There were relevant data of human MSCs in Results, but the culture method of human MSCs was not mentioned in Methods.
5.The abbreviations are different for the same noun. For example, in line 174 "canine umbilical cord-derived mesenchymal stromal cells (MSCs)" and in line 606 "canine umbilical cord-derived MSC (CUC-MSC)" and in line 178 "canine (K9, middle)". Those will confuse the reader. Please confirm that.
6. I suggest adding a negative control group in Figure 4B. Because human MSCs don't exist under normal conditions. What is the control for K9 EVs?
7. It is recommended summarizing each result rather than only presenting the data in Result.
8. Why did the authors test "Protein Content of EV Samples" and "Dot blot" in page 8?
9.Why choose canine MSC-EV in study? MSC-EV is not usually a candidate for cytotherapy. I suggested choosing human MSCs from different tissue sources to study.
10. The data did not provide strong evidences for procoagulant activity of umbilical cord-derived MSC-EVs, I suggest adding animal experiments to detect procoagulant activity of umbilical cord-derived MSC-EVs in vivo.
11. I suggest rewriting the Discussion according to the main ideas of the study. The Discussion in this manuscript is unfocused.
Author Response
Reviewer #2:
This study is interesting with clinical significance. MSC-EV is a popular candidate for cytotherapy. It is necessary to evaluate the therapeutic risk of MSC-EV.
The authors put forward a new point of view for the therapeutic risk of MSC-EV via procoagulant activity. The followings are some comments to the authors.
Comments:
- I suggest that specific data on the results should be added in the Abstract.
RESPONSE TO REVIEWER 2: Thank you for this suggestion.
>>> Line 18-9 of the Abstract was edited per your suggestion.
2.If abbreviation had been defined in the text when used for the first time, abbreviation is recommended in the text below. For example, in line 62 "Extracellular vesicles (EVs) ".
RESPONSE TO REVIEWER 2: Thank you for this suggestion.
>> Line 62 was corrected per your suggestion.
- Please define all abbreviations in the text when used for the first time. For example, line 100"UC”,and line 101"SEC".
RESPONSE TO REVIEWER 2: Thank you for catching this oversight.
>> Lines 100 and 101 were changed per your suggestion.
- There were relevant data of human MSCs in Results, but the culture method of human MSCs was not mentioned in Methods.
RESPONSE TO REVIEWER 2: Thank you for pointing out this oversight. We added a sentence referencing the culture and characterization of human umbilical cord mesenchymal stromal cells to the Methods section. The sentence goes just before the Design of Experiments section.
>>> Added to Ms before line 641: “…Human umbilical cord mesenchymal stromal cells were isolated, expanded and characterized as described in [98,99].”
Two references were added to Ms.
- The abbreviations are different for the same noun. For example, in line 174 "canine umbilical cord-derived mesenchymal stromal cells (MSCs)" and in line 606 "canine umbilical cord-derived MSC (CUC-MSC)" and in line 178 "canine (K9, middle)". Those will confuse the reader. Please confirm that.
RESPONSE TO REVIEWER 2: Thank you for pointing out the confusion. We corrected Figure 4’s caption per your suggestion.
>>> Figure 4 caption, between lines 172-193, changed as indicated:
“Figure 4: Procoagulant activity of MSCs and EVs. (A) Tissue factor (TF) was expressed by human and canine umbilical cord-derived mesenchymal stromal cells. Note that the staining intensity of human MSCs (bottom) appears to be higher than canine MSCs. Calibration bar = 400 µM. (B) The procoagulant activity (FXa generation) of human and canine extracellular vesicles (EVs) isolated from umbilical cord mesenchymal stromal cells (MSCs). No significant differences in procoagulant activity levels were found between human MSCs (positive control, left bar) and canine ( middle) or human EVs. Data from three human MSC lines at passages 4 to 6 and three human and canine EV samples. (C) Procoagulant activity (ng/mL of FXa generated) of EVs isolated from canine umbilical cord-MSCs by cell passage early (P2-P5) versus late (P9-P12) passage. No statistical differences were found, but the trend was for late-passage EVs to have higher levels of procoagulant activity. (D) No significant differences were found between the isolation method used: size exclusion chromatography- (SEC, white bars) versus ultracentrifugation- (UC, gray bars) based EV isolation. There was a trend for EVs isolated via UC to have a higher procoagulant activity level than SEC. Data from EVs were derived from six canine MSC lines, three in each isolation method. (E) Canine EVs (Canine EVs) had significantly higher procoagulant activity than the negative control (FX + FVIIa). However, when incubated with polyclonal anti-tissue factor antibody, FX-a generation was not inhibited in K9 EVs. (F) Human EVs had significantly more procoagulant activity than the negative control (FX + FVIIa). The procoagulant activity of human EVs was significantly inhibited by an anti-TF antibody (clone HTF-1 was previously shown to inhibit the function of TF). In contrast, and similar to canine EVs, procoagulant activity was not inhibited by the polyclonal TF antibody (far right). Data are presented as the average of three biological replicates ± standard deviation. Asterisk indicates p < 0.05. “
NOTE: Figure 4 was modified to match the caption.
- I suggest adding a negative control group in Figure 4B. Because human MSCs don't exist under normal conditions. What is the control for K9 EVs?
RESPONSE TO REVIEWER 2: Thank you for your input. The reviewer indicates that human MSCs don’t exist in normal conditions. We interpret this comment to mean that MSCs are artificial, i.e., attached primary fibroblastic cells created by in vitro expansion of umbilical cord tissue. We point out that all the work here was in vitro (artificial). Human MSCs were previously reported to have procoagulant activity [1-4], as were human MSC-EVs [5], as reviewed [6]. In this context, human MSCs were used as the positive control to compare human and canine MSC-EV’s procoagulant activity in Figure 4b.
NOTE: No changes to Ms.
- It is recommended summarizing each result rather than only presenting the data in Result.
RESPONSE TO REVIEWER 2: Thank you for your suggestion. We agree that sometimes papers are more readable if the data are summarized and briefly discussed as you proceed through the Results section. This approach can result in a short Discussion since much Discussion has moved to the Results. As you noticed, we opted to keep the Results short and to summarize and discuss each result in the Discussion section. We considered adding a summary for each result, but owing to the Design of Experiments approach we used, we thought that the Results section would become too long and confusing should we opt for the approach you suggested.
NOTE: No changes to Ms.
- Why did the authors test "Protein Content of EV Samples" and "Dot blot" in page 8?
RESPONSE TO REVIEWER 2: Thank you for the question. We evaluated the protein content of EV samples because we had previously used protein as a standard for dosing EVs when we isolated them by ultracentrifugation (UC) [7]. When we changed EV isolation methods to size exclusion chromatography (SEC), there was a significant reduction in the protein content of the isolated EVs. This is unsurprising since other laboratories have noted UC’s co-isolation of soluble protein. That is why we reported protein content. We provided dot blots to comply with the International Society of Extracellular Vesicle’s EV publication standards [8].
NOTE: No changes to Ms.
9.Why choose canine MSC-EV in study? MSC-EV is not usually a candidate for cytotherapy. I suggested choosing human MSCs from different tissue sources to study.
RESPONSE TO REVIEWER 2: Thank you for the question. The dog is an excellent large animal model for human disease. For example, the National Institutes of Health (NIH)’s National Cancer Institute has added the dog as a large model for human cancer to its five-year strategic plan. In the context of human inherited bleeding disorders or tissue factor physiology, the dog is an excellent large animal model of the human, too. We added a sentence to the Introduction to clarify this point.
>>> Added to Ms at line 81: “… Since the dog is considered an excellent large animal model for human inherited bleeding disorders and contributed to the understanding of human tissue factor and Factor VIIa complex [9,10], here, we evaluated canine MSC-derived EVs for TF-expression and procoagulant activity.”
NOTE: Two new references have been added.
- The data did not provide strong evidences for procoagulant activity of umbilical cord-derived MSC-EVs, I suggest adding animal experiments to detect procoagulant activity of umbilical cord-derived MSC-EVs in vivo.
RESPONSE TO REVIEWER 2: Thank you for the suggestion. We agree that it is difficult to put the in vitro work performed here into a context for understanding the thromboembolism risk. Our work, and Chance et al. [1], shows that MSC-EVs express tissue factor and are procoagulant. We agree that performing in vivo experiments would provide additional information about the procoagulant activity of MSC-EVs, but it would not change our conclusions.
>> Changes to Ms: As part of the major modification of the Discussion section requested, we discuss this point under the new section header “Limitations of the present work”.
“…Here, TF expression and FXa generation were demonstrated in vitro. Quantitation of TF expression using a TF standard protein curve was unsuccessful and required troubleshooting and optimization or using another quantification method, such as a Western blot. Quantification by Western blot was not pursued here due to the low protein content of the SEC-isolated samples. Therefore, a stronger case for patient risk could be generated by adding a quantitative assay, such as a fibrin generation assay or an intravascular thrombosis induction assay [11,12]. “
NOTE: Two new references have been added.
- I suggest rewriting the Discussion according to the main ideas of the study. The Discussion in this manuscript is unfocused.
RESPONSE TO REVIEWER 2: Thank you for your suggestion. We agree that the Discussion section was too long and that we could improve the take-home message by removing redundancy and working on the flow. The Discussion was trimmed by eliminating 185 lines, and these lines were moved to the Supplemental file, Discussion section. In addition, the Discussion was rewritten to improve flow and focus. Again, thank you for pointing out this important and much-needed change.
>>> Changes to Ms:
- Lines 387-572 were moved to the Supplemental file, Discussion section.
- One New heading was added to the Supplemental file, Discussion section.
- The entire Discussion was rewritten to tighten and improve flow and focus.
- Section headings were added to the Discussion to improve focus and flow.
References
- Chance, T.C.; Rathbone, C.R.; Kamucheka, R.M.; Peltier, G.C.; Cap, A.P.; Bynum, J.A. The effects of cell type and culture condition on the procoagulant activity of human mesenchymal stromal cell-derived extracellular vesicles. J Trauma Acute Care Surg 2019, 87, S74-S82, doi:10.1097/TA.0000000000002225.
- Christy, B.A.; Herzig, M.C.; Montgomery, R.K.; Delavan, C.; Bynum, J.A.; Reddoch, K.M.; Cap, A.P. Procoagulant activity of human mesenchymal stem cells. J Trauma Acute Care Surg 2017, 83, S164-S169, doi:10.1097/TA.0000000000001485.
- Gleeson, B.M.; Martin, K.; Ali, M.T.; Kumar, A.H.; Pillai, M.G.; Kumar, S.P.; O'Sullivan, J.F.; Whelan, D.; Stocca, A.; Khider, W.; et al. Bone Marrow-Derived Mesenchymal Stem Cells Have Innate Procoagulant Activity and Cause Microvascular Obstruction Following Intracoronary Delivery: Amelioration by Antithrombin Therapy. Stem Cells 2015, 33, 2726-2737, doi:10.1002/stem.2050.
- Liao, L.; Shi, B.; Chang, H.; Su, X.; Zhang, L.; Bi, C.; Shuai, Y.; Du, X.; Deng, Z.; Jin, Y. Heparin improves BMSC cell therapy: Anticoagulant treatment by heparin improves the safety and therapeutic effect of bone marrow-derived mesenchymal stem cell cytotherapy. Theranostics 2017, 7, 106-116, doi:10.7150/thno.16911.
- Silachev, D.N.; Goryunov, K.V.; Shpilyuk, M.A.; Beznoschenko, O.S.; Morozova, N.Y.; Kraevaya, E.E.; Popkov, V.A.; Pevzner, I.B.; Zorova, L.D.; Evtushenko, E.A.; et al. Effect of MSCs and MSC-Derived Extracellular Vesicles on Human Blood Coagulation. Cells 2019, 8, doi:10.3390/cells8030258.
- Moll, G.; Ankrum, J.A.; Olson, S.D.; Nolta, J.A. Improved MSC Minimal Criteria to Maximize Patient Safety: A Call to Embrace Tissue Factor and Hemocompatibility Assessment of MSC Products. Stem Cells Transl Med 2022, 11, 2-13, doi:10.1093/stcltm/szab005.
- Abello, J.; Nguyen, T.D.T.; Marasini, R.; Aryal, S.; Weiss, M.L. Biodistribution of gadolinium- and near infrared-labeled human umbilical cord mesenchymal stromal cell-derived exosomes in tumor bearing mice. Theranostics 2019, 9, 2325-2345, doi:10.7150/thno.30030.
- Thery, C.; Witwer, K.W.; Aikawa, E.; Alcaraz, M.J.; Anderson, J.D.; Andriantsitohaina, R.; Antoniou, A.; Arab, T.; Archer, F.; Atkin-Smith, G.K.; et al. Minimal information for studies of extracellular vesicles 2018 (MISEV2018): a position statement of the International Society for Extracellular Vesicles and update of the MISEV2014 guidelines. J Extracell Vesicles 2018, 7, 1535750, doi:10.1080/20013078.2018.1535750.
- Knudsen, T.; Olsen, O.H.; Petersen, L.C. Tissue factor and factor VIIa cross-species compatibility. Front Biosci (Landmark Ed) 2011, 16, 3196-3215, doi:10.2741/3906.
- Nichols, T.C.; Hough, C.; Agerso, H.; Ezban, M.; Lillicrap, D. Canine models of inherited bleeding disorders in the development of coagulation assays, novel protein replacement and gene therapies. J Thromb Haemost 2016, 14, 894-905, doi:10.1111/jth.13301.
- van Es, N.; Hisada, Y.; Di Nisio, M.; Cesarman, G.; Kleinjan, A.; Mahe, I.; Otten, H.M.; Kamphuisen, P.W.; Berckmans, R.J.; Buller, H.R.; et al. Extracellular vesicles exposing tissue factor for the prediction of venous thromboembolism in patients with cancer: A prospective cohort study. Thromb Res 2018, 166, 54-59, doi:10.1016/j.thromres.2018.04.009.
- Oeller, M.; Laner-Plamberger, S.; Hochmann, S.; Ketterl, N.; Feichtner, M.; Brachtl, G.; Hochreiter, A.; Scharler, C.; Bieler, L.; Romanelli, P.; et al. Selection of Tissue Factor-Deficient Cell Transplants as a Novel Strategy for Improving Hemocompatibility of Human Bone Marrow Stromal Cells. Theranostics 2018, 8, 1421-1434, doi:10.7150/thno.21906.

Round 2
Reviewer 2 Report
This manuscript can be accepted in present form.